# Play Smart, Be Smart? Effect of Cognitively Engaging Physical Activity Interventions on Executive Function among Children 4~12 Years Old: A Systematic Review and Meta-Analysis

**DOI:** 10.3390/brainsci12060762

**Published:** 2022-06-10

**Authors:** Wenjing Song, Leyi Feng, Junwei Wang, Feifei Ma, Jiebo Chen, Sha Qu, Dongmei Luo

**Affiliations:** 1School of Sports Science, Beijing Sport University, Beijing 100084, China; 2019112025@bsu.edu.cn (W.S.); 2019210154@bsu.edu.cn (L.F.); 2020110060@bsu.edu.cn (J.W.); maff520@sxu.edu.cn (F.M.); 1004320180026@bsu.edu.cn (J.C.); s447844724@126.com (S.Q.); 2Institute of Sport Medicine and Health, Chengdu Sport University, Chengdu 610041, China; 3Department of Physical Education, Shanxi University, Taiyuan 030006, China

**Keywords:** physical activity, cognitively engagement, executive function, children

## Abstract

This paper aims to collect a compendium of randomized controlled trials (RCTs) exploring the effects of cognitively engaging physical activity (PA) interventions (basketball and floorball) on various domain-specific executive functions (EFs) in children aged 4 to 12. Following the PRISMA principle, 11 articles (total sample size: 2053) were analyzed for effect size and moderating impact with Stata 13.0 software. Overall EFs (SMD = 0.21, 95% CI 0.10 to 0.32, *p* < 0.05), updating (SMD = 0.17, 95% CI 0.03 to 0.30, *p* < 0.05) and shifting (SMD = 0.32, 95% CI 0.02 to 0.61, *p* < 0.05) were enhanced by cognitively engaging PA interventions. Age and BMI were found to have no effect on overall EFs performance in Meta regression. Overall EFs performance was improved by interventions with a session length (≥35 min) (SMD = 0.30, 95 % CI 0.10 to 0.49, *p* = 0.033). The review suggests that despite the moderate effect sizes, cognitively engaging PA may be an effective approach to improving EFs in children aged 4 to 12, especially updating and shifting.

## 1. Introduction

Executive Functions (EFs) are a high-level cognitive process that controls and adjusts other cognitive processes when completing complex cognitive tasks [1]. According to research, EFs are comprised of three primary components: updating, shifting, and inhibition [2,3,4]. Early EFs have been shown to predict children’s physical and mental wellbeing [5], academic achievement, notably arithmetic and reading skills [6]. Conversely, children with decreased EFs (i.e., inhibition deficits) are more likely to have behavioral and emotional issues [7], placing their families under a lot of physical and emotional stress [8]. Indeed, EFs develop fast during childhood (particularly between the ages of 5 and 12 years), and the formation of EFs during this time is crucial for future success [9]. As a result, methods to safely and effectively increase EFs in children have become a research hotspot.

Physical activity (PA) has received a lot of attention as a way to improve children’s EFs. Intervention studies have recently revealed that not all types of PA are equally beneficial to cognition. Aside from quantitative factors (such exercise length and intensity) [10,11,12,13,14], qualitative factors (such as exercise type) have been proven to influence children’s EFs [11,15,16,17]. To date, one of the most commonly researched qualitative elements of numerous forms of PA is cognitive engagement (CE) [18]. CE is defined as the level of cognitive effort required to master difficult skills and is thought to be induced by increased cognitive demand [19]. Complex skills necessitate more engagement of the prefrontal structures, and brain structure changes must be measured [20]. Furthermore, the “cognitive stimulation hypothesis” offers a plausible explanation for the cognitive benefits gained from PA cognitive demands. Cognitively challenging workouts are thought to engage brain areas that control higher-order cognitive processes [21,22]. However, the results of a number of research works aiming at determining the effect of sustained cognitively engaging PA interventions on EFs were mixed, with some showing a favorable benefit [21,23] and others having no effect [24] or even negative effects [25]. It is possible that the discrepancies are related to variances in intensity, session, duration, or physical demands (e.g., exergame, dancing, jogging) and/or the measures of EFs examined (e.g., N-Back Test, Go/No-Go Task, and so on).

In conclusion, while many studies have focused on the effects of PA, particularly cognitively engaging PA, on EFs in various populations, no study has conducted a quantitative analysis of those results, i.e., no evidence of a quantitative relationship between cognitive engagement in physical activity and EFs has been provided. Furthermore, the “dose effect” of PA on EFs and their sub-components in diverse populations with different cognitive engagement modalities, intensity, and duration has not been determined. Although studies have shown that several moderator variables exist between PA and EFs (e.g., the intensity and length of PA intervention, age, BMI, and so on), it remains to be seen if those variables can play a significant moderator role. Furthermore, while some studies have found that PA has a favorable effect on at least one area of EFs in children (6 to 12 years old), it is unknown whether this effect exists in preschoolers (3 to 6 years old). This review will be focused on the following objectives: (1) examine the “dose effect” of cognitively engaging PA on EFs; (2) examine whether factors such as intensity, duration, population type, and EF sub-components regulate the effect of cognitively engaging PA on EFs, with the goal of providing a reference for further discussion of precise exercise programs.

## 2. Materials and Methods

### 2.1. Search Strategy

This systematic review and meta-analysis was rigorously carried out in compliance with the established criteria of the PRISMA guidelines [26,27], the Handbook of Cochrane Collaboration [28], and PROSPERO (Registration Number: CRD42022302944).

One investigator (W.S.) searched 4 databases: PubMed, Web of Science, PsycINFO, and SPORTDISCUS. In primary searches, there were no limits on date, gender, or language. The evaluation period began with the launch of each database and ended on 31 December 2021. The AND operator was used to join the four fundamental components of Mesh phrases and keywords: (1) physical activity (e.g., exercise, exergame, cognitively engaging PA, training, chronic exercise, aerobic exercise), (2) EFs (e.g., cognitive function, updating, shifting, inhibition), (3) child (e.g., preschool, pupil), and (4) randomized controlled trial (e.g., RCT, randomized controlled, Cluster RCT). We also looked at the references for prior systematic reviews and meta-analyses in this field.

### 2.2. Inclusion and Exclusion Criteria

Studies that met the following criteria were considered for inclusion: (1) Studies conducted physical activity with cognitive demanding elements that aims to promote EFs in children; (2) Including experimental (cognitively demanding PA intervention) and control groups (regular physical education lessons or aerobic exercise intervention); (3) Duration of intervention was more than four weeks (4) Studies with RCT or cluster RCT design, and the subjects are children (age 4 to 12 years).

Studies were excluded if they: (1) only studied the effect of acute exercise or combined with other interventions (e.g., dietary intervention); (2) complete text not available; and (3) animal experiment, meeting review, or non-experimental studies.

### 2.3. Collection of Studies

Duplicate entries from database and reference list searches were initially eliminated in EndNote (version X9; Clarivate Analytics; East Haven, CT, USA). After the initial exclusion, the authors (W.S. and L.F.) filtered titles and abstracts independently according to inclusion criteria. Finally, two authors (W.J. and L.F.) independently assessed the full-text articles, and any inconsistencies were reviewed with a third author (S.Q.) until agreement was obtained.

### 2.4. Data Extraction

The current study retrieved and summarized data from the included studies, including publication year, author, subject description, study design, intervention method, intervention time, and outcome variables (Table 1). Using the Cochrane Collaboration Handbook as a guide, the mean and SD values of the pre-to-post intervention difference were calculated [28].

According with Cochrane Collaboration Handbook [28], studies that reported mean and SD values of pre and post-intervention were first retrieved, and then effect size (ES) of each included research was computed using values between the intervention (cognitively demanding PA intervention) and control groups (regular physical education lessons or aerobic exercise intervention). The formulas for calculating mean and SD pre- to- post change values were as follows: ‘Mean change = Mean post-Mean pre’ and ‘SD change = SQRT [(SD pre^2^ + SD post^2^) − (2 × Corr × SDpre × SDpost)]’, in which the correlation coefficient (Corr) was set to 0.5. For studies that only reported standard errors and 95% confidence interval, SD values were obtained by the formula ‘SD = SE × SQRT(N)’, SD = SQRT (N) × [(UCI − LCI)/3.92] (U = upper CI, L = lower CI) [12]. Because of the difference of measurements and instruments between studies, the pooled ES was estimated by standardized mean difference (SMD). Small, moderate, and large effect sizes are represented by SMD of 0.2, 0.5, and 0.8 [29].

### 2.5. Assessment of Study Quality

The Cochrane risk of bias tool advised adopting a “risk of bias” approach to assess study quality [30]. Before the evaluation, two researchers systematically studied the Cochrane evaluation manual and randomly selected 5 articles using a computer random number generator for pre-evaluation to ensure that the two reviewers had a consistent understanding of the evaluation criteria. The official evaluation was conducted in three rounds: in the first round, two researchers independently evaluated those studies according to the criteria, then in the second and third rounds, the items inconsistent with the previous round were evaluated and discussed again to reach the final agreement. Risk of bias of studies were categorized as “low”, “high” or “unclear” based on the presence of seven processes (random sequence generation, allocation concealment, blinding of participants and personnel, blinding of outcome assessment, incomplete outcome data, selective reporting, and other biases) [31]. At the same time, data were imported into Rev Man 5.3 software for analysis and processing, and bias risk maps were drawn to visually display bias.

### 2.6. Statistical Analysis

Stata 13.0 (College Station, TX, USA) was used to conduct present meta-analysis. A fixed-effects (*p* > 0.1 for I^2^) or random-effects model (*p* ≤ 0.1 for I^2^) was used for pooling the outcomes of the included studies based on the heterogeneity among studies. There is non-negligible heterogeneity between studies, and it is reasonable to choose random-effects model for effect size evaluation. In addition, considering that the results of the random-effects model are more extendable and the effect of heterogeneous groups is inevitable in social science research, this study adopts the random-effects model in effect size evaluation based on previous practices [12].

Additional statistical analysis included: (1) When a three- or multi-arm design included both cognitively engaging PA settings, traditional PA settings, and other settings, only the cognitively engaging PA arm and traditional PA arm were extracted as intervention and control groups, respectively; (2) When multiple instruments were used to measure the same EFs domain, only the more commonly used one was included; (3) Only the outcome of the more-executive demanding condition was included when several results on a single cognitive task were provided (e.g., incongruent trials in Flanker task) [32].(4) When a follow-up measurement was conducted in a study, only the post-intervention result was included [32].

Subgroup analyses based on three core EFs domains (updating, shifting and inhibition) were conducted after the overall meta-analysis. Intervention duration (<10 weeks vs. ≥10 weeks), session length (<35 min vs. ≥35 min), frequency of intervention (<3 times/week vs. ≥3 times/week), intervention time per week (≥100 min vs. <100 min) and total intervention time (<1000 min vs. ≥1000 min) were examined by subgroup analyses as well. Meta-regressions based on continuous variables such as age and BMI were conducted [33].

## 3. Results

The four database searches yielded a total of 1552 articles (Figure 1). Of the 155 studies eligible for full-text assessment, 11 studies met the inclusion criteria after duplications were removed and titles were reviewed. Table 1 highlights the fundamental information, categories of studies, features of study objects, and details of extraction of outcome indicators of intervention measures from the included literature.

**Table 1 brainsci-12-00762-t001:** Characteristics of studies included in the meta-analysis.

Study	Design	Participants Characteristic			EF Variables	Instrument	Intervention/Duration	
		Mean Age (Years)	N	Male/Female			Experimental Group the Form of Interventions	Control Group the Form of Interventions
Nejati et al. [34]	RCT	9.43	26	0/100	Updating Shifting Inhibition	①②③	EXCIR sessions 40–50 min/session, 3 times/wk, 5 weeks e.g., Color Jumping To jump on a color cell in a table of colors given the meaning of some presented color words on the screen.	aerobic exercise sessions 40–50 min/session, 3 times/wk, 5 weeks aerobic exercise program without cognitive load, running.
Meijer et al. (1) [24]	Cluster RCT	9.1	441	230/221	Updating Inhibition	④⑤	cognitively engaging PA 30 min/session, 4 times/wk, 12 weeks team games or exercises that require complex coordination of movements, strategic play, cooperation between children, anticipating the behavior of teammates or opponents, and dealing with changing task demands, such as dodge ball, basketball	aerobic exercise sessions 30 min/session, 4 times/wk, 12 weeks The focus was on highly repetitive and automated exercises, such as circuit training, relay games, playing tag, and individual activities like running or doing squats.
Meijer et al. (2) [24]	Cluster RCT	9.1	650	232/418	Updating Inhibition	④⑤	cognitively engaging PA 30 min/session, 4 times/wk, 12 weeks	regular PE session 30 min/session, 2 times/wk, 12 weeks Children in the control group followed their regular physical education lessons
Schmidt et al. [35]	Cluster RCT	5.34	137	64/73	Updating Inhibition Shifting	①⑥⑦	combined physical and cognitive training 15 min/session, 4 times/wk, 12 weeks The games were conceptualized to require gross motor movements, which in turn should increase PA, e.g., One Lizard, two lizards	regular PE session 15 min/session, 4 times/wk, 12 weeks The control condition consisted of an active waiting-list group
Oppici et al. [36]	RCT	8.8	50	28/32	Updating Inhibition Shifting	⑦⑧⑨	high-cognitive PA 60 min/session, 2 times/wk, 7 weeks The dance lessons took place during the participants’ PE and sport classes	regular PE session 60 min/session, 2 times/wk, 7 weeks
Chien et al. [37]	RCT	12.1	84	52/32	Inhibition	⑩	combined games and object manipulation skills; 3 times/wk, 8 weeks Movement concepts and skills focused on the ability to move in various situations, respond to speed, direction and force of movements, and control body movements while jumping, throwing, catching, dribbling, kicking, or passing.	regular PE session 3 times/wk, 8 weeks session were designed to focus on sport skill development
Egger et al. [21]	Cluster RCT	7.95	96	42/54	Updating Inhibition Shifting	⑨⑪⑫	cognitively engaging PA 10 min/time, 2 times/wk, 20 weeks For example, children were standing in a circle and playing the game “Horserace”.	aerobic exercise sessions 10 min/time, 2 times/wk, 20 weeks This condition was designed to promote children’s aerobic fitness. Although it is not possible to exclude cognitive engagement entirely from long-term PA interventions, the attempt was made to choose exercises that had as little cognitive demand as possible.
Benzing et al. [38]	RCT	10.63	51	42/9	Updating Shifting Inhibition	⑬⑭⑮	“Shape up” game 30 min/session, 3 times/wk, 8 weeks “Beatmaster Training Quest”: It consists of different exercises such as: (A) “Waterfall Jump”: The player stands on the edge of a waterfall and has to jump onto oncoming pieces (footprints) of wood in order not to fall down. While the frequency, size and order of the footprints vary the player has to jump with one or two legs in order to hit the footprints.	Waiting-list control group.
Gao et al. [39]	RCT	4.72	32	16/16	Shifting	⑦	exergaming intervention, The intervention program requested children participate in home-based educational exergaming using the Leap TV gaming console for at least 30 min/session 5 times/week beyond their usual PA.	regular PE session 30 min/session 5 times/wk, 12 weeks The control condition asked children to maintain regular PA patterns without any exergaming gameplay, with parents advised to not change their children’s regular PA routine during their child(ren)’s time in this condition.
Crova et al. [40]	RCT	9.6	70	35/33	Updating Inhibition	⑯	The enhanced PE programme 21 weeks with one curricular PE class per week plus two additional hours of skill-based and tennis-specific training. The curricular programme consisted of only one PE class per week and was focused on the development of fundamental motor skills and coordinative abilities, bodily expression and deliberate play	Traditional PE programme, 21 weeks
Pesce et al. [41]	cluster RCT	N/A	460	232/228	Updating Inhibition	⑯	cognitively engaging PA employed in this intervention had characteristics of deliberate play and deliberate preparation 1 h/week, 6 months	Traditional PE 1 h/week, 6 months
Schmidt et al. (1) [23]	RCT	11.3	126	54/72	Updating Shifting Inhibition	①⑨	Combined high PA and cognitive engagement. This intervention consisted of specifically designed team games (football and basketball) tailored to challenge EFs. 45 min/session, 2 times/wk, 6 weeks	Aerobic Exercise. This condition consisted of different group-oriented and playful forms of aerobic exercises, whose main aim was to promote children’s aerobic fitness. 45 min/session, 2 times/wk, 6 weeks
Schmidt et al. (2) [23]	RCT	11.3	124	54/70	Updating Shifting Inhibition	①⑨	Combined high PA and cognitive engagement. This intervention consisted of specifically designed team games (football and basketball) tailored to challenge EFs. 45 min/session, 2 times/wk, 6 weeks.	combined low PA and cognitive engagement, according to the national curriculum for physical education 45 min/session, 2 times/wk ,6 weeks

Note: ① n-back task; ② WCST, Wisconsin Card Sorting Test; ③ Go/No-Go Task; ④ verbal working memory; ⑤ Motor inhibition efficiency; ⑥ Day-night task; ⑦ DCCS, Dimensional Change Card Sort; ⑧ 2-list; ⑨ Flanker test; ⑩ Stroop test; ⑪ Backwards Colour Recall task; ⑫ “mixed” block within the flanker task; ⑬ modified Simon Task; ⑭ modified Flanker task; ⑮ modified color span backwards task; ⑯ RNG, random number generation task; EXCIR, Exercise for Cognitive Improvement and Rehabilitation.

### 3.1. Study Characteristic

In total, 11 investigations with a total of 2176 people were included in this study, with 2053 subjects in the final data analysis, with mixed genders and ages ranging from 4 to 12 years. A total of seven RCTs [23,34,36,37,38,39,40] and four studies [21,24,35,41] were clustered RCTs. In total, seven studies were carried out in Europe, one in America, two in Asia, and one in Australia. There were eleven studies that looked at the effects of cognitively engaging PA on core EFs, with nine studies looking at updating, seven studies at shifting, and ten studies at inhibition. There were five studies with a duration of less than 10 weeks, only two studies with session length less than 15 min, and total dose of intervention ranged from 360 min to 2520 min. Intensities were not reported in Nejati [34], Schmidt [35], Oppici [36] and Gao’s [39] work, but were measured in other seven studies for manipulation check. Meijer [24] and Egger [21] reported proportion of MVPA which measured by ActiGraph GT3X+, others accessed intensity by average heart rate during intervention [35,37,38,40,41], in which two studies [40,41] reported proportion of MVPA based on HR. MVPA proportion varied from 32.0% [24] to 49.6% [40], and the average HR reported was among 131.9 bpm [41] to 157.9 bpm [37].

### 3.2. Methodological Evaluation of the Included Literature

The Cochrane Risk of Bias (ROB) method was used to assess the quality of the included studies. All eleven have a “low” ROB for random sequence creation. A total of four studies had “low” ROB for allocation concealment, one research had “high” ROB, and six studies had “unclear” ROB. The ROB of ten was “high”, while one was “low” in terms of blinding the participants and workers. In the ROB of outcome assessment blinding, insufficient outcome data, and selective reporting, all eleven studies received a “low” rating. Overall, all of the investigations were of high quality. As a result, no studies were ruled out for further investigation (Figure 2 and Figure 3).

### 3.3. Heterogeneity Test and Sensitivity Analysis

The findings revealed that cognitively engaging PA interventions enhanced overall EFs with considerable heterogeneity (I^2^ = 64.7%, *p* < 0.01), indicating that the heterogeneity is high. Differences in outcome markers were the main source of heterogeneity due to the data features of this included study, which was further validated by sensitivity analysis. Excluding each study had a rather consistent influence on the overall results, as shown in the figure. Sensitivity analyses of the included studies were undertaken on a study-by-study basis, as shown in the figure. The leave-one-out sensitivity analysis identified two studies [34,41] as substantial contributors to the high heterogeneity, but the overall effect size change after exclusion was remained within the 95% CI, therefore no additional analysis of the excluded literature was conducted.

### 3.4. Effects of Cognitively Engaging PA on EFs

Because of the inter-study heterogeneity, the total effect size was computed with a random-effects model, and the weight of each study was modified utilizing the D-L approach, as shown in Figure 4. Overall EFs had a pooled SMD of 0.21 (95 % CI 0.10 to 0.32, I^2^ = 64.7%, *p* < 0.05), The SMD was 0.17 (95% CI 0.03 to 0.30, I^2^ = 44.6%, *p* = 0.054) for updating, 0.32 (95% CI 0.02 to 0.61, I^2^ = 67.5%, *p* < 0.01) for shifting; and 0.18 (95% CI −0.01 to 0.37, I^2^ = 74.2%, *p* < 0.01) for inhibition. As a result, the substantial variation for core EFs highlights the necessity of considering underlying characteristics when examining the impacts of cognitively engaging PA.

### 3.5. Moderator Analysis

Age, BMI, duration of intervention, frequency, and other confounding factors were all taken into account in the studies. Thus, those variables were included by pre-sent moderator analysis. Age and BMI were conducted by a meta-regression, and no significance was reached. Meanwhile, other moderators were conducted by subgroup analyses (Table 2), in which session length showed significant moderate effect. Heterogeneity between subgroups presented that longer session length (≥35 min).

## 4. Discussion

### 4.1. The Overall Effect of Cognitively Engaging PA on EFs

This study intergraded the available literature in this field of inquiry to quantitatively examine the effect of chronic cognitively engaging PA interventions (typically over 4 weeks) on core EFs (updating, shifting, and inhibition) in children aged 4–12 years. We found a small but significant positive effect size of cognitively engaging PA interventions on overall EFs, updating specifically. In Meta regression, age and BMI were conducted by a meta-regression, and no significance was reached. Interventions with a session length ≥35 min, improved overall EFs performance. No other moderator was found to have an effect.

### 4.2. Comparisons with Previous Studies

The results of this study showed that cognitively engaged PA has a favorable effect on overall EF performance in children aged 4 to 12. As a result, our work is a helpful addition to three previously published systematic reviews [12,42,43]. Chronic exercise interventions had a minor but substantial effect on overall EFs and inhibitory control specifically, according to a review that comprised 19 RCTs studies [12]. Another systematic review [36] found that long-term physical activity improved inhibitory control slightly (SMD = 0.2, 95% CI 0.03–0.37; *p* = 0.021) [42]. Furthermore, De Greeff [43] found that longitudinal PA programs have a positive effect on EFs (SMD = 0.24; 95% CI 0.09–0.39; 12 studies). Notably, only three studies (n = 3) directly compared the effects of aerobic and cognitively demanding PA (n = 3) and found that the combination of aerobic and cognitively demanding PA had a greater effect than aerobic exercise without or with low cognitive engagement. Although PA appears to have a wide range of favorable effects on a variety of cognitive processes, the benefits of a cognitively engaging PA intervention appear to be greater for Efs [44]. There is, however, no agreement on which cognitive activities are more susceptible to PA therapies.

### 4.3. Analysis of Regulatory Variables between Cognitively Engaging PA and EFs

Based on the existing research experience [12] and the characteristics of this study, the moderating variables between PA and EFs were divided into age, BMI, duration of intervention, frequency, session length, intervention dose per week and total dose of intervention. In the heterogeneity test of the total effect size, *p* < 0.01, I^2^ = 64.7% > 50%, Specific discussions are as follows.

Age [45] and BMI [46] are important regulatory factors between PA and EFs. However, in our findings, the moderating effects of age and BMI are not obvious. The study found that older children may benefit more from physical activities with complex rules [23]. On the one hand, children of different ages of the nervous system maturity, developmental condition, hormone level, able to complete the action, understand the rules exist great differences, so in order to optimize the cognitive participation, researchers in determining intervention plan needs to be age factors into consideration, think carefully about the children’s development [47,48,49]; On the other hand, physical activity that consistently challenges children’s cognitive abilities has the greatest effect on improving Efs [50,51,52,53,54]. In terms of BMI, previous studies have shown that obese children may benefit more from long-term physical activity than their normal-weight peers. Gustafson [55] believes that obesity causes subclinical inflammatory changes in the brain, including changes in blood vessels and demyelination of white matter, resulting in cognitive impairment. Nevertheless, the meta-regression revealed that EFs score of overweight and normal-weight children were not statistically significant after the intervention. The possible reason is that the sample size of overweight children is small [37,40].

In addition to Age and BMI, session length, intervention dose per week and total dose of intervention were also moderator on the effect of cognitively engaging PA on EFs. Specifically speaking, interventions with session length (≥35 min), seemed to have no noticeable effect on EFs. The reasons for the findings are not clear. The effect of long-term PA on EFs was better than that of short-term PA, and it was a medium effect [56]. As can be observed, the findings of this investigation are largely consistent with those of previous studies [12,33,57].

### 4.4. Cognitively Engaging PA Changes the Underlying Mechanism of EFs in Children

According to new research, cognitively engaging PA is more likely to offer greater cognitive benefits than non-cognitively engaging PA. One possible explanation is that cognitively engaging PA activates the same frontal-dependent neural networks that are activated when EFs are activated. Increased activation of these brain networks after a bout of PA may result in more efficient neural functioning during subsequent cognitive activities, resulting in improved performance, the neuronal network directly recruited by cognitively engaging PA is the same as EFs, suggesting that this could be one mechanism. Increased neural network activation increased neuronal functioning, which could contribute to higher cognitive performance [11,22,58]. The co-activation and interconnection of the brain regions linked to cognition and movement, may give synergistic effects when cognitive and PA are combined. When the job is demanding, novel, needs focus, and the required response is unpredictable and quick, neuronal co-activation is at its peak. Improved cognitive performance is the outcome of co-activation elicited by some environmental factors that are stimulated by cognitively stimulating PA [59,60,61].

On the other hand, combining cognitive and physical activities may produce synergistic effects due to co-activation and inter-connectedness of the neural areas associated with cognition and movement (referring broadly to the prefrontal cortex and the cerebellum, respectively) [58,62]. This neural co-activation is strongest when the task is demanding, novel, requires concentration, and when the required response is unpredictable and quick. Therefore, cognitively engaging physical activities may stimulate the necessary contextual parameters to elicit co-activation resulting in enhanced cognitive performance [11]. In contrast to PA-only interventions cognitively engaging PA interventions appear to have stronger favorable impacts on EFs [63]. The reason for this is that higher cognitively engaging PA requires more complex cognitive engagement in order to cooperate with partners, anticipate companion and opponent behavior, adapt movement strategies to changing task requirements, a process that necessitates more cognitive and social interaction, and the need to mobilize the neural circuits associated with executive functions to participate in [64,65]. The prefrontal region’s activity level has increased, which is equivalent to what the brain nervous system requires for children to complete executive function activities, hence executive function has improved [66,67].

### 4.5. Strengths and Limitations

The following are some of the current study’s advantages. First, the analysis process included only high-quality experimental studies; no observational studies were included, ensuring the current result’s credibility. Second, we measured the impact of PA interventions on children’s cognitive capacities in our research. Third, crucial modifiers were considered in the study, such as the type of intervention, the characteristics of the exercise task, and the duration of the intervention were all taken into account in the study.

Meanwhile, there are certain limitations to this analysis that may restrict the trustworthiness of this result in some ways. To begin with, only publications written and published in English are included, which means that some high-quality studies written in other languages may be overlooked. Second, the included studies used various study designs, eligibility criteria, follow-up durations, and intervention strategies, which could contribute to inconsistent results. Furthermore, there are no universal criteria for measuring the cognitive engagement of exercise intervention among the studies included. Furthermore, the Meta-analysis resulted in the statistical method of inter-study integration of effect sizes, which could lead to misunderstanding of results due to the small number of studies, therefore the result should be read cautiously.

## 5. Conclusions

Despite the small impact sizes, this research found that cognitively engaging PA could be a potential method to increase various elements of EFs, particularly updating and shift. We should encourage children to engage in more physical activity, particularly physical activity with higher cognitive demands, because it is safe, low-cost, and beneficial for both physiological and cognitive health.

## Figures and Tables

**Figure 1 brainsci-12-00762-f001:**
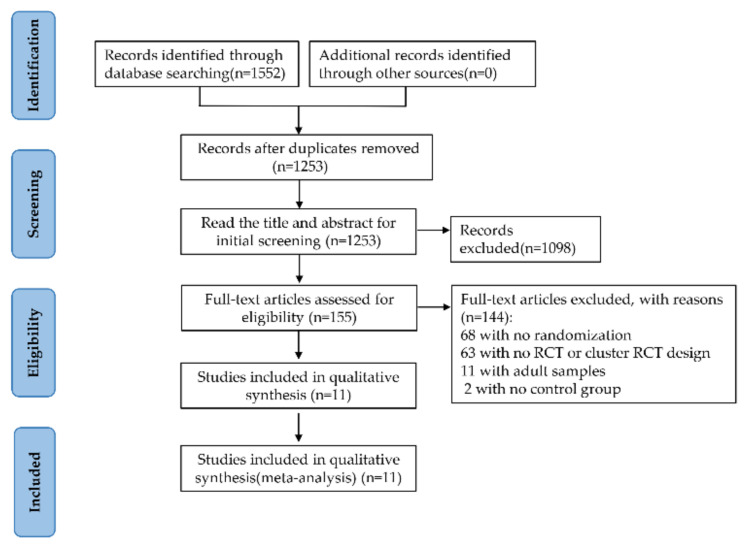
Flowchart of Literature Search and Study Selection.

**Figure 2 brainsci-12-00762-f002:**
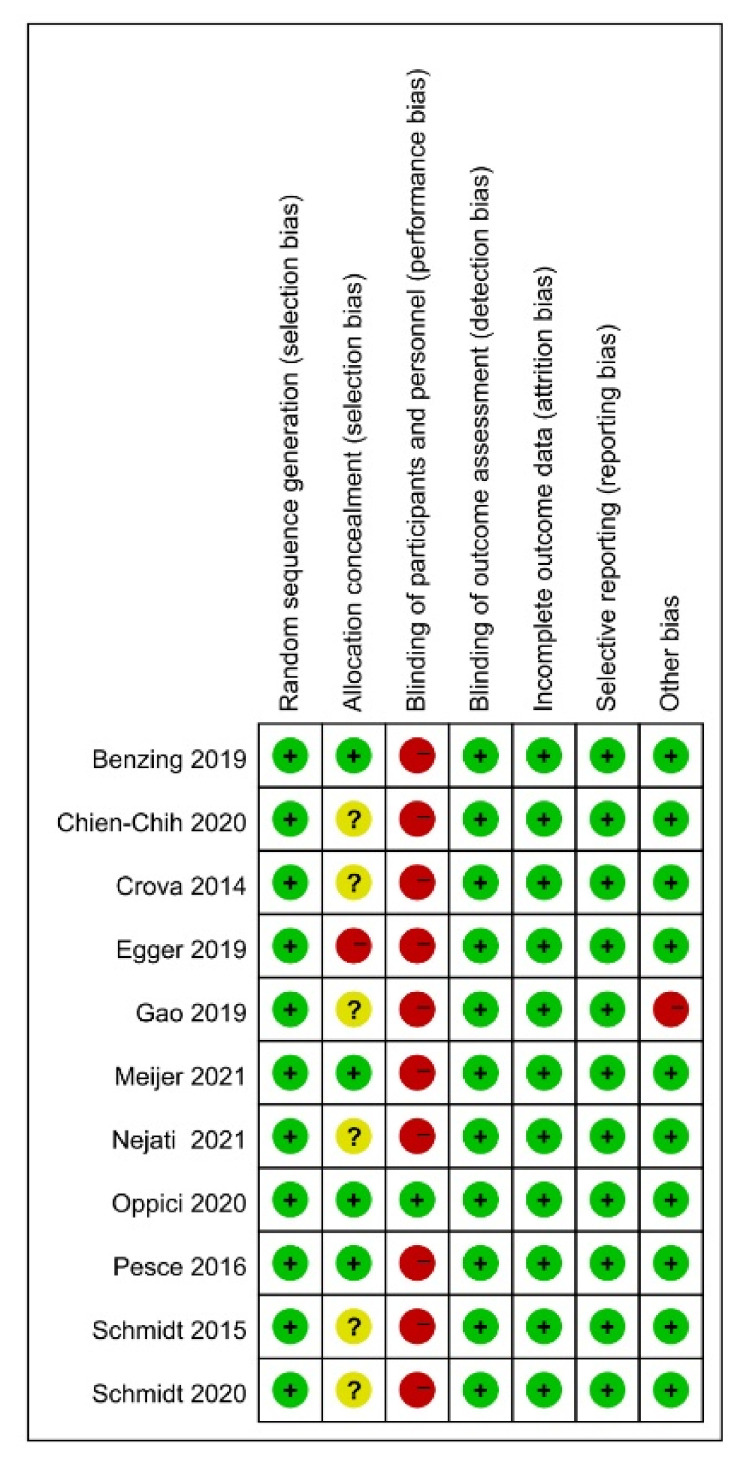
Quality evaluation results of included studies’ risk of bias (ROB). Ref. [23] ROB levels: low (green or “+”), unclear (yellow or “?”), and high (red or “−”).

**Figure 3 brainsci-12-00762-f003:**
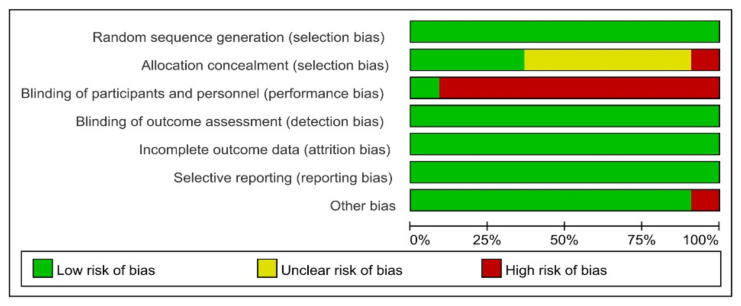
Quality evaluation results of included studies’ risk of bias (ROB). ROB levels: low (green), unclear (yellow), and high (red).

**Figure 4 brainsci-12-00762-f004:**
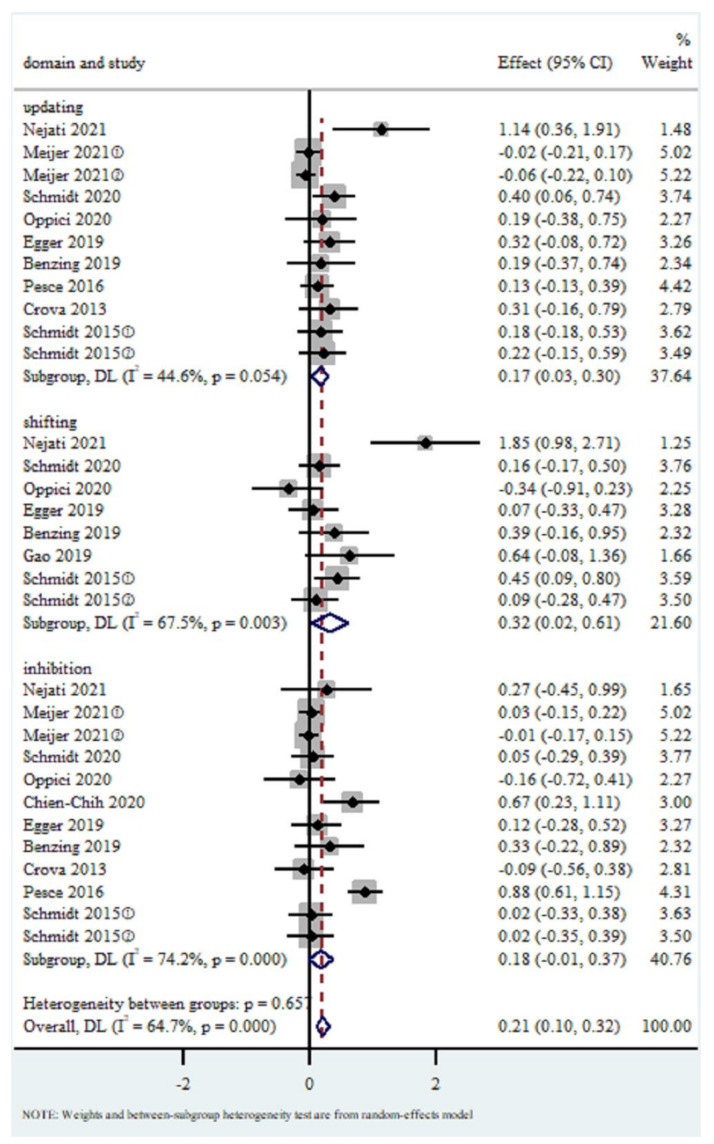
Forest plot for a meta-analysis of the effects of cognitively engaging PA on different EF domains.

**Table 2 brainsci-12-00762-t002:** Moderator analysis of cognitively engaging PA and EFs.

Categorical Variables	Level	No. of Studies	Cohen’s d	95%CI	I^2^%	Heterogeneity between Subgroups		
						Q	d.f.	*p* Value
Session length(min)	<35	14	0.07	−0.01 to 0.15	10.4	4.55	1	0.033
	≥35	17	0.30	0.10 to 0.49	71.5			
frequency	<3	13	0.18	−0.01 to 0.36	65.7	0.12	1	0.725
	≥3	19	0.22	0.09 to 0.35	62.5			
Dose (min)/week	<100	14	0.26	0.11 to 0.40	55.8	0.74	1	0.390
	≥100	17	0.16	−0.56 to 0.38	63.4			
Duration (week)	<10	19	0.25	0.11 to 0.40	46.1	0.71	1	0.401
	≥10	12	0.16	0.00 to 0.32	71.0			
Total dose (min)	<1000	18	0.23	0.08 to 0.37	45.7	0.08	1	0.774
	≥1000	13	0.19	0.03 to 0.36	76.5			
Continuous variables	Level	No. of studies	β	95%CI	I^2^%	Adjusted R^2^%	*p* value	
BMI	15–25	28	0.03	−0.03 to 0.10	56.9	1.59	0.40	
Age	4–12	31	0.03	0.04 to 0.09	63.8	5.48	0.27	

## Data Availability

The data used to support the findings of this study are included within the article.

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
