# Peer review of "Play Smart, Be Smart? Effect of Cognitively Engaging Physical Activity Interventions on Executive Function among Children 4~12 Years Old: A Systematic Review and Meta-Analysis"

_brainsci, 2022, doi:10.3390/brainsci12060762_

Round 1

Reviewer 1 Report

Thank for the opportunity to review “Play Smart, Be Smart? Effect of Cognitively Engaging Physical Activity Interventions on Executive Function among Children 3~18 Years Old: A Systematic Review and Meta-Analysis.” The authors summarize 11 randomized controlled trials conducted in children and adolescents that reported both cognitively challenging and ‘traditional’ physical activity interventions. Overall, I think this an interesting and relevant topic in pediatric public health. However, the actual aims of the review were not consistently addressed through the manuscript and my strong concern is that the approach did not properly/specifically address the aims.

GENERAL COMMENTS

The aims were not consistently described throughout the article and interpretations/conclusion often did not seem specific to what was actually analyzed here. In the abstract, the aim was “to collect a compendium of randomized controlled trials (RCTs) exploring the effects of cognitively engaging physical activity (PA) interventions on various domain-specific executive functions (EFs) in children aged 3 to 18.” As written, this seems as if it would just be a descriptive review of RCTS that have used cognitively engaging PA interventions. In the introduction, the authors were more specific with “ (1) examine the "dose effect" of  cognitively engaging PA on EFs; (2) examine whether factors such as intensity, duration, population type, and EF sub-components regulate the effect of cognitively engaging PA on EFs, with the goal of providing a reference for further discussion of precise exercise programs.” It was not mention a comparison group and reads as though they will tease apart which components were most effective in improving EFs. After reading the results/summary table, I don't think modalities/components are clearly described/teased out. The PROSPERO objective states: "How does physical activity intervention with high cognitive engagement affect children's executive function compared to physical activity with low cognitive engagement?" This seems to be what this review actually explored. But, the aims described in the manuscript do not match up with the approach.

Can we truly look at dose effect/response without comparing to inactive/cognitively challenging activities?

I think it is important for the authors to make clear how they are defining "cognitively engaging PA". Without a clear description, one could think of most structured activities in early childhood as cognitively engaging given the emphasis on developing new or refining gross and fine motor skills. Most PA interventions in preschool incorporate gross motor skill development and many games will involve EFs (e.g., Freeze Dance). A 'tag' game in older children can be cognitively challenging (reaction time, change in speed, etc.). Then there are games that explicitly require EFs.

It would also be important to motivate/justify including the wide age range encompassing early childhood to adolescents, given that PA intervention modalities/approaches differ across ages and EF has develops across these ages.

Overall, the manuscript needs some attention to editing. There were some incomplete sentences, typos, and awkward phrasing.

SPECIFIC COMMENTS

Abstract, Line 15: Suggest "Following" rather than "According"

Line 33: Can the authors explain this? What is meant by low? Perhaps, discussing briefly the development of EFs across early childhood through middle childhood would be helpful here. It would also be helpful to mention other correlates and determinants of EFs across childhood and transition to the focus of physical activity.

Line 44: Links/mechanisms for why these types of activities vs non-cognitively challenging (which should also be defined) would be as beneficial or better should be discussed as well.

Line 59: Please be more specific about what is meant here - there are many recent reviews exploring PA and cognitive measures in early childhood. While findings are inconsistent in this age, there are RCT studies. And activity is "prescribed" differently in these ages (hard to define specific doses) - but that could also be said for older children.

For example:

Carson V, Hunter S, Kuzik N, et al. Systematic review of physical activity and cognitive development in early childhood. J Sci Med Sport. 2016;19(7):573–578.  

Zeng N, Ayyub M, Sun H, Wen X, Xiang P, Gao Z. Effects of physical activity on motor skills and cognitive development in early childhood: a systematic review. Biomed Biomed Res Int. 2017

St. Laurent CW, Burkart S, Andre C, Spencer RMC. Physical Activity, Fitness, School, Readiness, and Cognition in Early Childhood: A Systematic Review. J Phys Act Heal. 2021;18(8):1004–13.

Lines 60-63: I think the aim here would need to be more specific as it appears that the included articles had active controls - so it appears to more about exploring whether they have similar or better effects than traditional PA and how do the 'prescriptions' differ?

Line 63: This is not a practical implication of this work. Precise exercise programs are not generally/broadly focused on for children or the public health message. Interventions focus more on PA enhancement/opportunities (throughout day). Suggest reframing/considering this.

Line 81: As noted earlier, it is important to define what is meant here. And how was it determined that the research group used these element specific to target EFs?

Line 83: What were the comparators? Please add sections/details that follow the PRISMA sections/categories.

Line 123: Please name these processes so readers understand what categories contributed to bias risk.

Line 136: Why? This isn't explicitly motivated in the introduction and as noted above, subsections to describe exposure/intervention and comparators (as done for PROSPERO) would be important to include. According to the registered protocol, comparators were "Regular PA setting, with low cognitive engagement (i.e., repetitive and nonadaptive PA)" - PE lessons do not necessarily fall into that category (and should not if following current PE teaching practices).

Table 1. With a small sample size, it would be good to see more about the content/context of the experimental activities - perhaps in a supplemental table - particularly as the authors are trying to get information about specific dose/programming. In discussion, I hope authors mention that based on some comparison groups, there could be "bleeding" of the in target exposure. Regular PE classes could still have cognitively challenging elements. Also, in the Meijer (2) study for example, it also looks like the experimental group had double the volume - so can't tease apart effects of just changing the types of activity. Schmidt study - Why is this one included if the experimental group had sedentary training and not PA? Oppici study - what is meant by low-cognitive PA?

Line 194: Relative to all comparison groups? It would be more meaningful to understand it was the same or better than 'traditional' PA (however that is defined), as well as non-active groups (i.e., sedentary cognitively challenging activities) or no intervention. Here it does look like all comparison groups were active as well.

Line 233: Again, to be more accurate this really should be stated in comparison to PA programs that has less cognitively challenging activities (although this really needs to be defined as to how that was determined). I think dosage should be tempered given the approach here. Also missing is summary of modality - that was mentioned in aim but not really teased apart in results or here in summary.

Line 309: Not sure I agree this should be listed as a strength as you could not look at these questions with observational designs. The other strengths as well seem subjective - I would recommend referring to the approach rather than the strength of the research aims.

Line 320: Agree with this, but also makes it important to clearly explain and motivate how this was defined/parameterized in current review.

Line 232: This also supports my caution that the interpretation/summary of these findings should be clear and accurately described based on what this review did and found.

Line 327: This conclusion does not appear matched to the aims.

Line 328: One of the stated goals was to inform PA programming. This conclusion suggests that PA should promoted - which is already agreed upon and supported. Please highlight what these findings are contributing.

Author Response

Reply: Thanks to your valuable comments. To make this study more specifically to its aim, we already have made some modifications refer to your comments, including adjustments in comparators and study inclusion.
We have modified our article to be consistent with our aim, which is to make a comparation between the effect of PA intervention with cognition engagement intervention and PA intervention without emphasized cognitive elements. There were several studies that don’t correspond with the aim above have been excluded for more specific analysis to the cognitive element, then we revised the analysis and results. Along with other modifications refer to your important comments, we’ve highlighted them in our revised manuscript, you can check it in detail.
Once again, we appreciate your comments. 

Reviewer 2 Report

Song and colleagues reviewed studies investigating the effects of cognitively engaging physical activity on executive function in children 3-18 years old. Please see my comments and suggestions below.

Title

  1. Authors should clarify physical activity and exercise. Since intervention studies are covered here, exercise makes more sense.
  2. One of my biggest concerns lies in 3-18 years old children. Can we call those with age 18 children? There are tremendous changes in the neural development from 3-18 and it is very likely that the effects of intervention on executive function differ between 3-18. Therefore, I have a concern of drawing a generalized conclusion of intervention on executive function using an age range of 3-18.

Abstract

  1. Line 14 – examples are needed for the cognitively engaging PA interventions
  2. Lines 21 and 22 – I think there are typos in 95% CI 17 and 11.
  3. Line 25 – Cognitive engagement should be grammatically correct for the key word.

Introduction

  1. Line 32 – adult is not relevant to this study so I recommend to remove this.
  2. Lines 33 – reference(s) is needed to support this sentence.
  3. Line 34 – it sounds weird that EF can be trained by PA and other interventions. PA and other interventions may boost the improvement but, not necessarily ‘train’ EF.
  4. Line 43 – the transition is not smooth. Authors should include justification about examining PA with cognitively engaging rather than PA itself.
  5. Line 63 – As I mentioned earlier PA and exercise are different and should not be used interchangeably.
  6. Rationale for studying 3-18 years old seems not clear. Using 3-18 years old as well as potentially different effects of intervention on EFs due to neural development should be discussed.

Methods

  1. Lines 135 and136 – What are the examples of cognitively engaging PA settings, traditional PA settings, and other settings? Need clarifications.
  2. Line 138 – what is the definition of ‘more commonly used one’? This should be crystal clear given this study is a review study.
  3. Lines 138 to 140 – This sentence needs clarification. It sounds unclear to me.
  4. Line 144 – how the cutoffs of the intervention duration, session length, frequency of intervention, intervention time per week, and total intervention time were determined?

Results

  1. Table 1 – not sure if country column is necessary given that other more important information are missing.
  2. Table 1 – since this is a review study, I highly encourage the authors to clarify the form of interventions for readers. For example, which form of aerobic exercise? Was it treadmill or cycling? Which form of cognitively engaging PA? That way, the readers don’t need to check the reference to see which form of interventions were used. Overall, the table should contain specific description about the intervention, which I think is very important.

Discussion

  1. Line 231 – should it be 3-18 years?
  2. Line 261 – What is the age range of older children? If it is over 15, it is hard to say that they are children. They are adolescent.
  3. Lines 269 to 271 – Did Gustafson study children? The neural mechanism may differ between children and adults so if it was studied on adults, it may not applicable in children.
  4. Line 274 – What does ‘data error’ mean?
  5. Line 286 – what is the example of ‘Highly cognitively engaging PA, complex PA in the background environment’?
  6. Line 290 – which neural network are authors referring to?
  7. Line 295 to 296 – need a reference for this sentence.
  8. Line 298 – the second paragraph of the 4.4 section mainly discusses acute effects of exercise on EFs, which is different from the effects induced by chronic intervention. What is the process of accumulated effects indued by multiple bout of exercise on the neural networks engaging in EFs?
  9. Line 299 – the third paragraph needs clarification. The authors mention team sports but team sports are different from PA with cognitive engagement.
  10. Lines 305 to 307 – which neuroimaging technique was used to observe the prefrontal activity?

Author Response

Thanks to your valuable comments. To make this study more specifically to its aim, we already have made some modifications refer to your comments, including adjustments in comparators and study inclusion.                                                          

Title

Comment :One of my biggest concerns lies in 3-18 years old children. Can we call those with age 18 children? There are tremendous changes in the neural development from 3-18 and it is very likely that the effects of intervention on executive function differ between 3-18. Therefore, I have a concern of drawing a generalized conclusion of intervention on executive function using an age range of 3-18.

Reply :Thanks for your comment, we’ve altered the title as Play Smart, Be Smart? Effect of Cognitively Engaging Physical Activity Interventions on Executive Function among Children 4~12 Years Old: A Systematic Review and Meta-Analysis

Abstract

Comment :. Line 14 – examples are needed for the cognitively engaging PA interventions

Comment : .Lines 21 and 22 – I think there are typos in 95% CI 17 and 11.

Reply :Dear expert, I checked the data carefully and found no errors of 17 and 11

Comment :Line 25 Cognitive engagement should be grammatically correct for the key word.

Reply :Thanks for your suggestion, we’ve changed phrase “Cognitively Engagement” to “cognitive engagement”

Introduction

Comment Line 32 – adult is not relevant to this study so I recommend to remove this.

Reply :Thanks for recommending, we’ve removed that.

Comment Lines 33 – reference(s) is needed to support this sentence.

Thanks for your comment, we’ve added the following references:

  1. Stern, A.; Pollak, Y.; Bonne, O.; Malik, E.; Maeir, A. The Relationship Between Executive Functions and Quality of Life in Adults With ADHD. J Atten Disord 2017, 21, 323-330, doi:10.1177/1087054713504133.
  2. Contreras-Osorio, F.; Campos-Jara, C.; Martínez-Salazar, C.; Chirosa-Ríos, L.; Martínez-García, D. Effects of Sport-Based Interventions on Children's Executive Function: A Systematic Review and Meta-Analysis. Brain Sci 2021, 11, doi:10.3390/brainsci11060755.
  3. Diamond, A.; Ling, D.S. Conclusions about interventions, programs, and approaches for improving executive functions that appear justified and those that, despite much hype, do not. Dev Cogn Neurosci 2016, 18, 34-48, doi:10.1016/j.dcn.2015.11.005.

Comment Line 34 – it sounds weird that EF can be trained by PA and other interventions. PA and other interventions may boost the improvement but, not necessarily ‘train’ EF.

Reply: We agree with the expert's suggestions, the word “train” have been replaced.

Comment Line 43 – the transition is not smooth. Authors should include justification about examining PA with cognitively engaging rather than PA itself.

Reply : We’ve added a transition paragraph as follows: “Conversely, children with EFs deficits(e.g., inhibition deficits) are more likely to have behavioral and emotional issues [7], which would put their families under a lot of physical and emotional stress [8]. Indeed, EFs develop fast during childhood (particularly be-tween the ages of 5 and 12 years), and the formation of EFs during this time is crucial for future success [9]. As a result, how to facilitate EFs efficiently in childhood has become a research hotspot.”

Results

Comment Table 1 – not sure if country column is necessary given that other more important information are missing.

Comment Table 1 – since this is a review study, I highly encourage the authors to clarify the form of interventions for readers. For example, which form of aerobic exercise? Was it treadmill or cycling? Which form of cognitively engaging PA? That way, the readers don’t need to check the reference to see which form of interventions were used. Overall, the table should contain specific description about the intervention, which I think is very important.

Reply to comments for table 1 :Thanks for these comments. We’ve modified Table 1 in the manuscript.

Discussion

Comment :Line 231 – should it be 3-18 years?

Line 261 – What is the age range of older children? If it is over 15, it is hard to say that they are children. They are adolescent.

Reply:Thanks for this comment, the age range of included studies was only up to 12 years old, so we used the term “children” to refer them.

  1. Lines 269 to 271 – Did Gustafson study children? The neural mechanism may differ between children and adults so if it was studied on adults, it may not applicable in children.
  2. Line 274 – What does ‘data error’ mean?

Reply: Nevertheless, the meta-regression revealed that EFs score of overweight and nor-mal-weight children were not statistically significant after the intervention. The possi-ble reason is that the sample size of overweight children is small

  1. Line 286 – what is the example of ‘Highly cognitively engaging PA, complex PA in the background environment’?

Reply: The rationale for the additive cognitive effects derived from cognition engagement was based primarily on the contextual-interference effect hypothesis and its application to motor skill acquisition and rehabilitation research.

  1. Line 290 – which neural network are authors referring to?

Reply: One potential mechanism is that cognitively engaging PA directly recruits the same frontal-dependent neural networks used when EFs are demanded. Increased activation of these neural networks during a bout of PA may lead to more efficient neural functioning during cognitive tasks that follow this exercise resulting in enhanced performance.

  1. Line 295 to 296 – need a reference for this sentence.
  2. Line 298 – the second paragraph of the 4.4 section mainly discusses acute effects of exercise on EFs, which is different from the effects induced by chronic intervention. What is the process of accumulated effects indued by multiple bout of exercise on the neural networks engaging in EFs?
  3. Line 299 – the third paragraph needs clarification. The authors mention team sports but team sports are different from PA with cognitive engagement.
  4. Lines 305 to 307 – which neuroimaging technique was used to observe the prefrontal activity?

Reply: Dear Expert, the author made significant changes to the discussion as suggested, as detailed in the manuscript                                                                 Once again, we appreciate your comments. 

Round 2

Reviewer 2 Report

The authors did a good job addressing the concerns with a quick turnaround. I endorse the publication of this manuscript.